# Transapical Transcatheter Aortic Valve Replacement: A Real-World Early and Mid-Term Outcome of a Third-Level Centre

**DOI:** 10.3390/jcm11144158

**Published:** 2022-07-18

**Authors:** Alessandra Francica, Filippo Tonelli, Alberto Saran, Gabriele Pesarini, Igor Vendramin, Rocco Tabbì, Cecilia Rossetti, Giovanni Battista Luciani, Flavio L. Ribichini, Francesco Onorati

**Affiliations:** 1Department of Surgery, Dentistry, Paediatrics, and Gynaecology, Division of Cardiac Surgery, University of Verona Medical School, 37126 Verona, Italy; filipp0tonelli92@gmail.com (F.T.); saran.alberto@gmail.com (A.S.); rocco.tabbi@aovr.veneto.it (R.T.); rossetticeci93@gmail.com (C.R.); giovanni.luciani@univr.it (G.B.L.); francesco.onorati@univr.it (F.O.); 2Department of Medicine, Division of Cardiology, University of Verona Medical School, 37126 Verona, Italy; gabriele.pesarini@aovr.veneto.it (G.P.); flavio.ribichini@univr.it (F.L.R.); 3Cardiothoracic Department, Azienda Sanitaria Universitaria Friuli Centrale, 33100 Udine, Italy; igor.vendramin@asuiud.sanita.fvg.it

**Keywords:** transapical transchateter aortic valve replacement, transchateter aortic valve implantation, severe aortic valve stenosis, transapical valve-in-valve implantation, Heart-Team

## Abstract

Background: Transapical transcatheter aortic valve replacement (TA-TAVR) is generally considered to be associated with higher morbidity compared with transfemoral-TAVR. However, TA-TAVR remains a feasible alternative for patients who are unsuitable for TF-TAVR. It has been shown that outcomes after TAVR are linked to the operator’s expertise. Therefore, the purpose of this study is to report short- and mid-term outcomes after TA-TAVR performed by an expert Heart-Team of a third-level centre. Methods: From 2015 to 2022, 154 consecutive patients underwent TA-TAVR. The outcomes were analysed according to the VARC-3 criteria. Kaplan–Meier curves were estimated for major clinical events at mid-term follow-up. Results: The mean age of the population was 79.3 years and the STS risk-score of mortality was 4.2 ± 3.6%. Periprocedural mortality was 1.9%. Acute kidney injury and prolonged ventilation occurred in 1.9%. Incidence of stroke was 0.6%. Pacemaker implantation rate was 1.9%. Freedom from cardiovascular mortality was 75.7%, and 60.2% at 3 and 5 years. Freedom from stroke was 92.3% and 88.9% at 3 and 5 years, respectively; freedom from endocarditis was 94.4% and 90.8% at 3 and 5 years, respectively. Conclusion: TA-TAVR may be considered a safe and effective alternative approach in patients unsuitable for TF-TAVR, especially when performed by a proficient Heart-Team.

## 1. Introduction

Nowadays transcatheter aortic valve replacement (TAVR) is recognized as the gold-standard alternative treatment for severe aortic stenosis in elderly patients at high risk for surgery [1,2]. The rapid technological refinements of TAVR devices led to improved clinical outcomes. In this scenario, the transfemoral (TF-TAVR) access is currently regarded as the first-line approach [1,2]. In the last few years, different percutaneous vascular accesses (i.e., trans-carotid, trans-subclavian, trans-axillary) have gained great popularity as possible routes for TAVR, even if data are still limited [3,4,5]. Even though its use is decreasing, TA-TAVR still represents one of the alternative approaches supported by the greatest worldwide experience [6,7,8,9]. Moreover, the recent literature suggests that improved outcomes may be derived from more controlled TAVR programs in high-volume centres [10,11], and that the operator’s experience is the key to achieve and maintain the most favourable clinical results—especially in higher-risk subjects [11,12]. At our Institution, the interdisciplinary Heart-Team started the TA-TAVR program in 2015, and since then a large number of patients have been recruited for TA procedures—both TAVR and Valve-in Valve (VIV) implantations. The aim of this study is to report on early- and mid-term clinical outcomes of TA procedures.

## 2. Materials and Methods

### 2.1. Study Population

From March 2015 to April 2022, all patients affected by severe aortic stenosis undergoing transapical transcatheter aortic procedures at the Division of Cardiac Surgery of University Hospital in Verona were enrolled. The balloon expandable SAPIEN devices (Edwards Lifesciences, Irvine, CA, USA) were used for both TAVR and Valve-in-Valve (ViV) procedures in the aortic position. The interdisciplinary Heart-Team selected patients for the TA approach according to the latest European and American guidelines for the management of valvular heart diseases [1,2]. The appropriate prosthetic size was determined on computed tomography findings. All patients underwent general anesthesia, and all the procedures were carried out in the catheterization laboratory by an expert cardiac surgeon together with a proficient interventional cardiologist that took care of the actual valve positioning and deployment.

Pre-procedural, intra-procedural, and post-procedural data were retrospectively collected in a dedicated and anonymized database. Follow-up clinical data were gathered by querying the Electronic Clinical Chart (retrieving data from the Regional Health Database) or the patients. The clinical follow-up was complete for 100% of the patients. Informed consent was obtained from all of the patients.

### 2.2. End-Points

The primary endpoint of the study was to assess periprocedural mortality defined as death occurring ≤30 days after the index procedure or >30 days but during the index hospitalization according to the latest Valve Academic Research Consortium 3 (VARC-3) criteria [13].

Secondary endpoints included the following major post-operative events.

-Stroke was defined as an overt central nervous system injury according to STS [14] and VARC-3 definitions [13].-According to the “Clinical Practice Guidelines for Acute Kidney Injury 2012 [15], stage 3 of acute kidney injury (AKI) was diagnosed if at least one of the following criteria was present: increase in serum creatinine >300% (>3.0 × increase) within 7 days compared with baseline or serum creatinine of 4.0 mg/dL (354 mmol/L) with an acute increase of ≥0.5 mg/dL (≥44 mmol/L). The post-operative requirement of renal replacement therapy was also assessed.-Prolonged ventilation was defined if mechanical ventilation >24 h was needed according to STS definition [14].- Post-procedural bleeding was classified as Type 1 (minor), Type 2 (major), Type 3 (life-threatening), and Type 4 (leading to death) bleeding according to VARC-3 and BARC criteria [13]-Vascular and access-related non-vascular complications were classified as major or minor complications based on VARC-3 definition [13]-Other acute procedural and technical valve-related complications, including conversion to surgery, unplanned use of mechanical circulatory support, implantation of multiple (>1) valves during the index hospitalization because of valve malposition, and thrombosis and paravalvular regurgitation, were also in line with VARC-3 criteria [13].-New cardiac conduction disturbances and arrhythmias, including atrial fibrillation, atrioventricular block, or other abnormalities requiring permanent pacemaker and/or implantable cardioverter-defibrillator implantation, were defined according to VARC-3 criteria [13].-Periprocedural myocardial infarction definition was according to the “Fourth Universal Definition of Myocardial Infarction (2018)” [16]-Length of intensive care unit (ICU) stay was registered.

Finally, mid-term outcomes were estimated. Therefore, the overall survival as well as freedom from cardiovascular mortality [13], from stroke, from endocarditis, and from re-hospitalization related to cardiovascular causes [13] were estimated at 1, 3. and 5 years after the TA procedure.

### 2.3. Statistical Analysis

Descriptive statistics were used to analyze data. Categorical variables are presented as absolute values and frequencies (%) and continuous variables are presented as means with standard deviations (SDs). Kaplan–Meier curves were estimated for freedom of main outcome events. The statistical analysis was performed using SPSS Version 26.0 (Armonk, NY, IBM Corp.). No adjustments for multiple testing were performed. All the analyses were exploratory.

## 3. Results

### 3.1. Baseline Characteristics and Predictive Risk-Scores

A total of 154 consecutive patients underwent TA procedures because of severe native aortic valve stenosis (*n* = 144) or severe degeneration of a previous aortic prosthesis (*n* = 10). The mean age of the population was 79.4 years and 88 patients (57.1%) were males. Most of the patients (80.5%) had pre-operative frail conditions and were affected by severe peripheral artery disease (70.6%) or “porcelain” aorta (49.4%). Eighteen (17.5%) patients had prior CABG surgery, while ten patients (6.6%) underwent previous surgical aortic valve replacement. The mean EuroSCORE II was 6.7 ± 5.2% and the mean STS-score risk of mortality was 4.2 ± 3.6%; the STS risk-score for mortality and morbidity was 16.8 ± 8.8%. All the STS risk-scores for each post-procedural complication were calculated and shown in Table 1.

### 3.2. Periprocedural Outcomes

The number of TA procedures progressively increased over the period of the study (Figure 1) and no intraprocedural deaths occurred. Eight patients underwent concomitant Percutaneous Transluminal Coronary Angioplasty (PTCA); three PTCA were urgently performed because of acute relevant coronary obstructions that intraprocedurally supervened. At the end of valve deployment, the intra-operative echocardiographic assessment showed mild and moderate PVL in 14.9% and 1.3% of cases, respectively. Severe PVL was never recorded. Intraprocedural variables are displayed in Table 2.

Periprocedural mortality was 1.9% (*n*= 3) (see Table 3 and Figure 2). One patient developed early thrombosis after the ViV procedure, while the other two deaths were related to a hemorrhagic stroke and mesenteric ischemia. AKI stage-3 occurred in 1.9%, whereas no patients needed dialysis. One patient (0.6%) developed stroke while two patients (1.3%) required prolonged ventilation (see Table 3 and Figure 3).

Seven patients (4.5%) suffered from minor vascular complications (i.e., perivascular hematoma or femoral pseudoaneurism). One patient needed open surgical revision because of cardiac tamponade. Three patients (1.9%) required post-procedural PM implantation. The mean length of ICU stay was 1.6 ± 1.3 days. Most of the patients were discharged in the NYHA class I-II (93.5%). All peri-procedural outcomes are displayed in Table 3.

### 3.3. Mid-Term Outcomes

The mean follow-up time was 4.5 ± 0.3 years. Overall survival was 89.6%, 68.5%, and 48.3% at 1, 3, and 5 years (Figure 4), whereas freedom from cardiovascular mortality was 90.9%, 75.7%, and 60.2% at 1, 3, and 5 years after the procedure (Figure 5A). Freedom from stroke was 97.1%, 92.3%, and 88.9% at 1, 3, and 5 years (Figure 5B). Freedom from endocarditis was 96.9%, 94.4%, and 90.8% at 1, 3, and 5 years (Figure 5C). In particular, endocarditis occurred in seven patients: two patients underwent open cardiac surgery with good results, while only one patient benefited from antibiotic treatment. Freedom from re-hospitalization related to cardiovascular causes was 90.6%, 76.7%, and 65.7% at 1, 3, and 5 years after the index hospitalization (Figure 5D).

## 4. Discussion

Our experience shows excellent outcomes in patients undergoing TA-TAVR. The in-hospital mortality rate (1.9%) was lower than estimated by both the pre-operative EuroSCORE II (6.7 ± 5.2%) and STS-score risk of mortality (4.2 ± 3.6%) (see Figure 2). It was also lower than the mean rates previously reported for TA procedures [17,18,19]. Blackstone E.H. et al. [18] reported 7.8% of in-hospital mortality for TA-TAVR in the sub-analysis of the first PARTNER trial. Other observational studies from TAVR registries reported an incidence of 30-day mortality ranging from 4% to 8% with contradictory findings when compared with a TF approach [8,9,10,19,20,21]. Our excellent results support a recent study [12] that highlights the relevance of the operators’ experience in achieving and maintaining optimal clinical outcome after transcatheter procedures. Of note, the incidence of major co-morbidities resulted in predictive STS risk-scores that were estimated lower for each post-operative complication (See Figure 3). In particular, the low rate of stroke (0.6%) probably is derived from avoiding the manipulation of multiple devices and catheters into the aortic arch, as shown by previous investigations [9,18,19,20,21]. Similarly, both the incidence of post-operative AKI and prolonged ventilation was lower than predicted by STS risk-scores (see Figure 3). These data suggest that the greater invasiveness of the TA procedure does not necessarily affect post-operative outcomes. As expected, we registered a few minor vascular complications and only one minor access-related complication, thereby supporting previous studies that highlighted the higher incidence of major vascular complications in TF-TAVR compared to TA-TAVR [8,18,19,20,21,22]. Furthermore, several studies comparing the two access routes reported a higher percentage of permanent PM implantation after TF-TAVR [8,18,19,20,21,22]. Of note, post-operative PM implantation was necessary in only 1.9% of our patients. This result probably depends on the easier valve crossing and excellent anterograde controllability through the apex of the left ventricle. These characteristics, along with the SAPIEN 3 and SAPIEN 3 Ultra delivery systems [23,24], might also explain the low incidence of moderate PVLs (1.3%) and the absence of severe PVLs after valve implantation.

To the best of our knowledge, this is one of the largest third-level Centre reports on the 5-year outcomes after TA-TAVR. A previous multicentre Italian study [25] reported a 1-year survival of 81.7% and 3-year survival of 67.6%, while freedom from cardiovascular mortality turned out to be 91.2% and 83.1% at 1 and 3 years, respectively. Our results support previous findings and also show satisfying outcomes up to 5 years after TA-TAVR.

Our experience suggests that TA-TAVR might be a safe alternative for candidates who are not suitable for surgery or TF-TAVR. However, it could be speculated that our promising results may depend on the Heart-Team’s expertise, as suggested by recent analyses that remarked the inverse association between mortality in both centre and single-operator case volume [10,11,12]. This probably contributes to the different outcomes of TA access when compared to the other approaches and might also explain the differences between our results and those reported previously.

### Limitations

The main limitation of the study relates to the retrospective nature of our investigation. However, our results come from a real-world daily practice of a third–level Centre. Furthermore, to the best of our knowledge, this is the first TA-TAVR analysis reporting early- and mid-term outcomes of TA-TAVR according to the VARC-3 criteria.

## 5. Conclusions

TA-TAVR can be considered as a safe and effective alternative approach in high-risk patients that are unsuitable for TF-TAVR, especially when carried out in a third-level Centre by an expert Heart-Team.

## Figures and Tables

**Figure 1 jcm-11-04158-f001:**
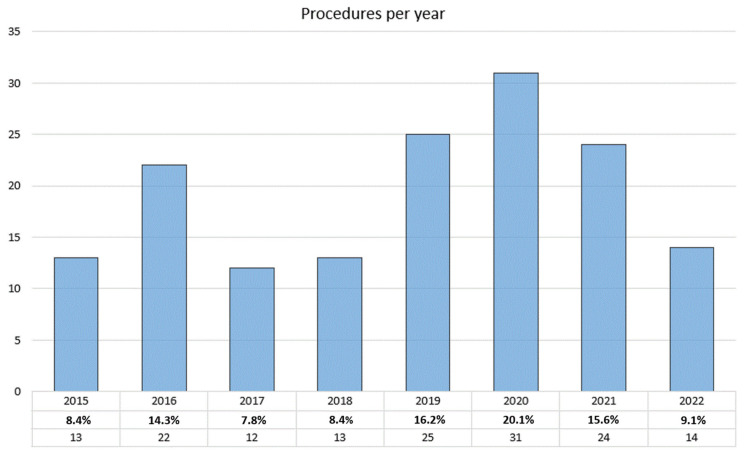
Transapical aortic procedures performed annually from January 2015 to April 2022.

**Figure 2 jcm-11-04158-f002:**
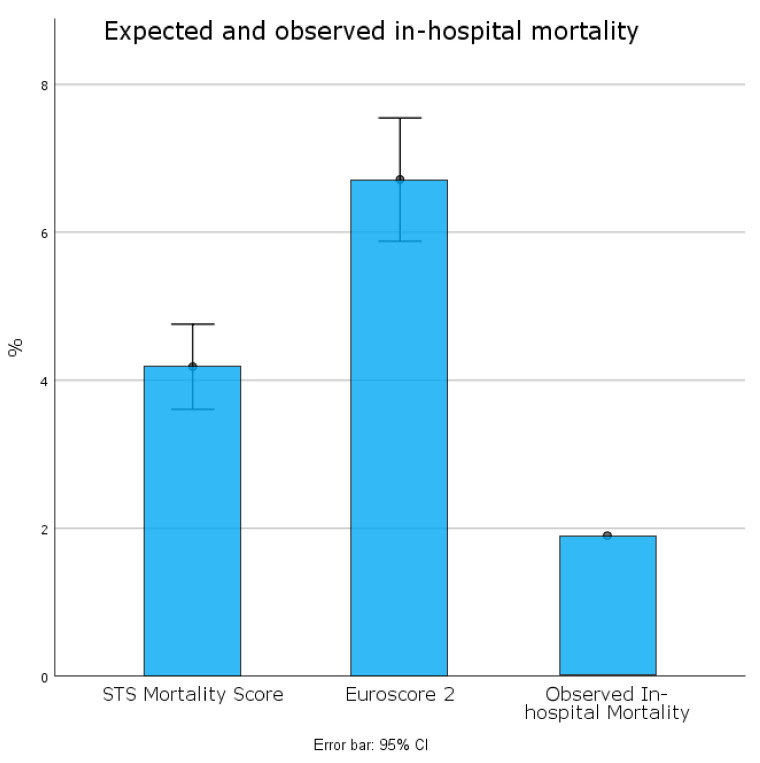
Expected and observed periprocedural mortality.

**Figure 3 jcm-11-04158-f003:**
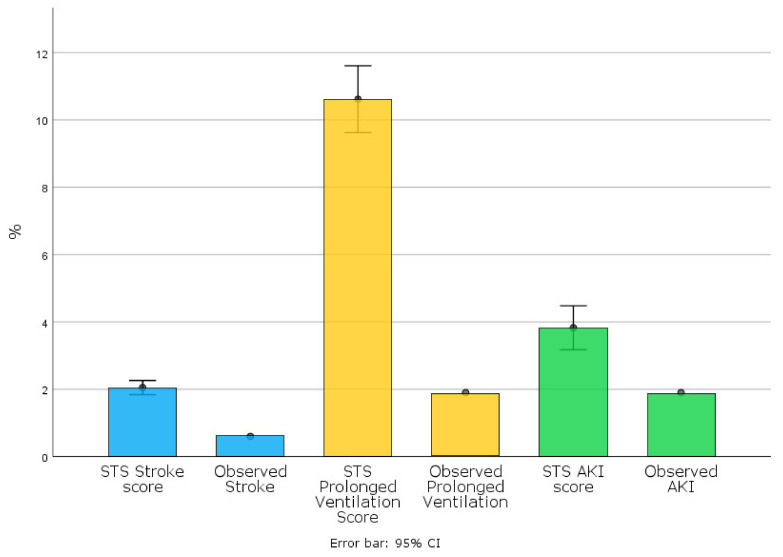
Expected and observed periprocedural complications.

**Figure 4 jcm-11-04158-f004:**
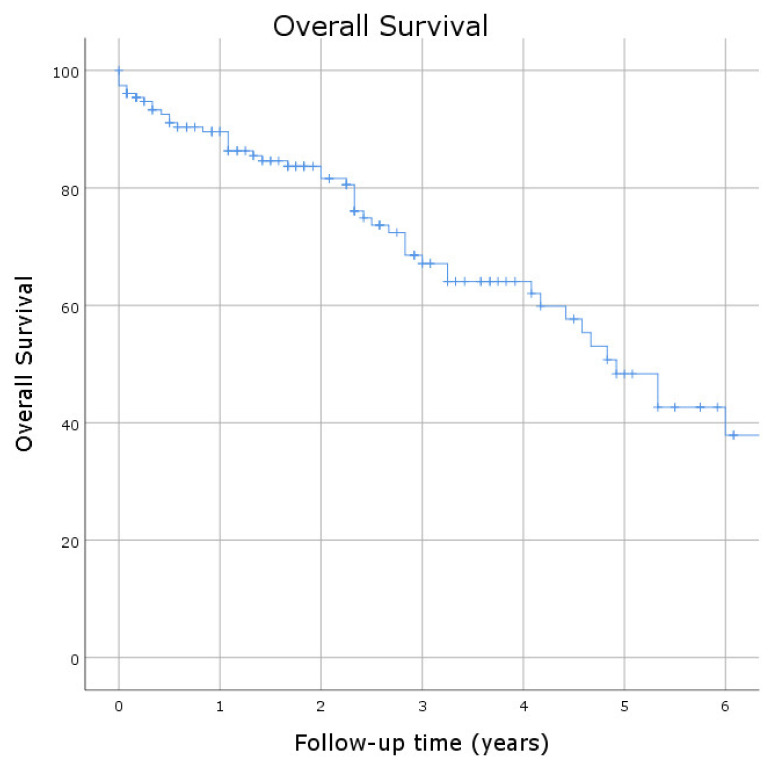
Kaplan–Meier curve estimated for the overall survival at mid-term follow-up.

**Figure 5 jcm-11-04158-f005:**
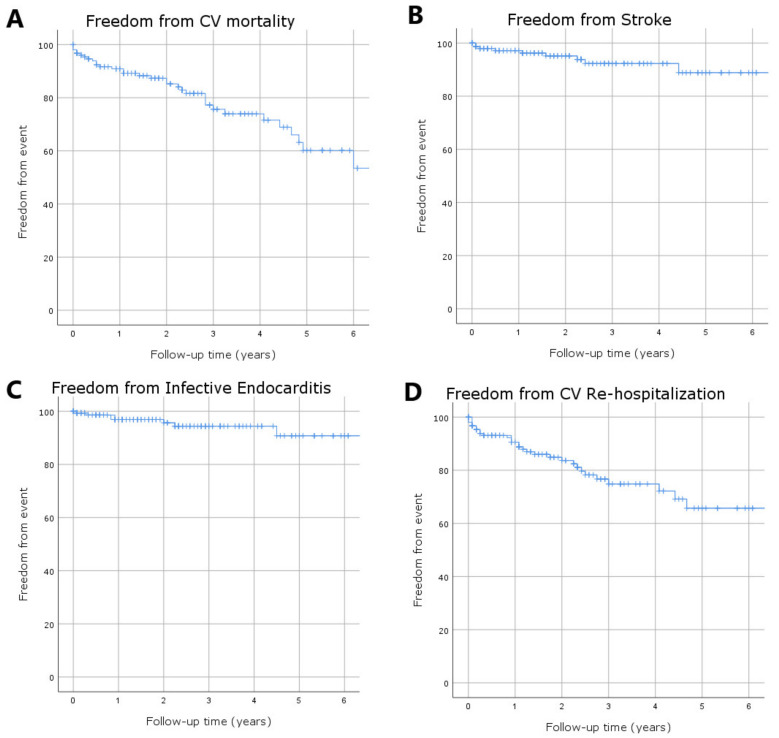
Kaplan–Meier curves estimated for freedom from major clinical events at mid-term follow-up: (**A**) freedom from cardiovascular (CV) mortality; (**B**) freedom from stroke; (**C**) freedom from endocarditis; and (**D**) freedom from re-hospitalization due to CV causes.

**Table 1 jcm-11-04158-t001:** Baseline variables.

Baseline Characteristics *n* (%); m (SD)	Patients Undergoing TA Aortic Procedures *n* = 154
Age, years	79.4 (5.7)
BMI, kg/m^2^	27.2 (4.7)
>75 years old	130 (85.5)
Males	88 (57.1)
Hypertension	134 (87)
Dyslipidemia	92 (49.7)
Diabetes Mellitus ID	12 (7.8)
Chronic lung disease >mild	65 (42.6)
Dialysis	2 (1.3)
Creatinine, umol/L	103.5 (56.7)
Tabagism	
former	18 (11.9)
active	60 (39.1)
Atrial fibrillation	
persistent	60 (32.4)
paroxysmal	15 (8.1)
Pacemaker	12 (7.8)
Carotid stenosis > 50%	74 (48.7)
History of cerebrovascular disease	
Stroke	16 (10.4)
TIA	6 (3.9)
Peripheral artery disease	109 (70.6)
Coronary artery disease	78 (50.6)
Frailty *	136 (80.5)
Cirrhosis	12 (7.8)
Porcelain aorta	76 (49.4)
Previuos PCI	27 (17.5)
Previous CABG	18 (11.7)
Previuos valvular surgery	10 (6.6)
NYHA class III–IV	103 (66.8)
LVEF, %	57.3 (10.4)
Euroscore II, %	6.7 (5.2)
STS Risk of Mortality, %	4.2 (3.6)
STS Renal Failure, %	3.8 (4.04)
STS Prolonged Ventilation, %	10.6 (6.2)
STS Stroke, %	2.05 (1.3)
STS Morbidity and Mortality, %	16.8 (8.8)

* Katz Index of Independence in Activities of Daily Living > 3 [17] Legend: BMI, Body Mass Index; ID, insulin dependent; LVEF: Left Ventricular Ejection Fraction; TIA, Transient Ischemic Attack; PCI, Percutaneous Coronary Intervention; NYHA, New York Heart Association.

**Table 2 jcm-11-04158-t002:** Intraprocedural variables.

Procedural Variables *n* (%); m (SD)	Patients Undergoing TA Aortic Procedures *n* = 154
TAVR	144(77.8)
VIV	10 (11.8)
Intraprocedural PTCA	8 (5.2)
Staged PTCA	11 (7.1)
Procedural time, min	113.5 (31.4)
Prosthetic type	
Edwards SAPIEN XT	3 (1.9)
Edwards SAPIEN 3	100 (64.9)
Edwards SAPIEN 3 ultra	51 (33.2)
Intraprocedural mortality	-
Paravalvular leak	
mild	23 (14.9)
moderate	2 (1.3)
severe	-

Legend: TAVR, Transcatheter aortic valve replacement; VIV, Valve-in-Valve; PTCA, Percutaneous Transluminal Coronary Angioplasty.

**Table 3 jcm-11-04158-t003:** Periprocedural outcomes.

Postprocedural Outcomes *n* (%); m (SD)	Patients Undergoing TA Aortic Procedures *n* = 154
Periprocedural mortality	3 (1.9)
Bleeding	
Type 1	2 (1.3)
Type 2	2 (1.3)
Type 3	1 (0.6)
Type 4	-
Vascular complications	
Major	-
Minor	7 (4.5)
Access-related non-vascular complications	
Major	-
Minor	1 (0.6)
Cardiac tamponade requiring surgical revision	1 (0.6)
Valve malposition	-
Valve thrombosis	1 (0.5)
AKI stage 3	3 (1.9)
New dialysis	-
Peak of creatinine, umol/L	119.8 (75.6)
Neurological events	
Stroke	1 (0.6)
TIA	-
New onset atrial fibrillation	42 (27.3)
PM implantation	3 (1.9)
Coronary obstruction	3 (1.9)
Periprocedural myocardial infarction	3 (1.9)
Length of ICU stay, days	1.61 (1.3)
Prolongued ventilation (>24 h)	3 (1.9)
Pneumonia	5 (3.2)
Wound infection	2 (1.3)
LVEF, %	55.3 (8.4)
NYHA class at discharge	
I	71 (46.1)
II	73 (47.4)
III	7 (4.5)
IV	-

Legend: AKI, Acute Kidney Injury; TIA, Transient Ischemic Attack; PM, Pacemaker; ICU, Intensive Care Unit; LVEF: Left Ventricular Ejection Fraction; NYHA, New York Heart Association.

## Data Availability

The data presented in this study are available in the manuscript.

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
