# Peer review of "Transapical Transcatheter Aortic Valve Replacement: A Real-World Early and Mid-Term Outcome of a Third-Level Centre"

_jcm, 2022, doi:10.3390/jcm11144158_

Round 1

Reviewer 1 Report

Francica A, Tonelli F et al investigated 185 patients undergoing transapical procedures (TAVR as well as mitral valve procedures) in this retrospective single-center study and were able to show an excellent short- and long-term outcome, furthermore they compared the secondary endpoints to the predicted risk (EuroScore, STS score). 

So far, non-transfemoral TAVR may only be considered in inoperable patients unsuitable for TF-TAVR therefore this study may offer a new perspective on TA-TAVR.

Certain points of the study are of notice: 

-) the wording of the title "transapical transcatheter aortic valve replacement" seems misleading considering that also patients with mitral valve procedures were included. I would suggest to exclude the patients undergoing mitral valve intervention and only keep the patients with aortic valve intervention to create a more homogenous group. Alternatively, I would at least suggest to show separate statistics of the aortic and mitral valve group (baseline, outcome, mortality/morbidity).

-) the definition of acute kidney injury lacks "initiation of renal replacement therapy"  - according to the RIFLE criteria. If it just not mentioned in the methods please adapt the methods accordingly, otherwise please also adapt the results. 

-) As secondary endpoint, permanent stroke is mentioned - was a neurological follow up performed? Since "permanent" stroke is not a definition of VARC3 I would suggest to change the wording. 

-) In the text as well as in the tables are some minor and major language errors. 

Reviewer 2 Report

This is an interesting study on apical TAVR procedures, which shows good results. It is however known that the traditional risk scores are not appropriate for TAVI and it is therefore questionable to compare the results to these scores. On the other hand, good alternatives for the traditional scores are lacking so far.

In my opinion the apical is a safe method as well and by doing it experience can be maintained for future procedures like TMVI and  ascending aorta stunting etcetera. However, the transfemoral and subclavian routes show superior results and the number of patients in need of other accesses declines.

I have no further suggestions for improvement.

Reviewer 3 Report

This is a well written manuscript addressing an important topic. Dr. Francica et al., reported their own short- and mid-term outcomes after transapical transcatheter aortic vlave replacement (TA-TAVR). As the authors mentioned, transfemoral (TF) TAVR is the first-line approach for TAVR. However, sometimes the femoral access is not suitable due to severe peripheral arterial disease.   One of the alternative approaches in this case is the TA approach. The results from the authors are very good and help propagate for the TA approach as an alternative to TF approach in selected patients. The study is well designed , endpoints are well defined, and the statistical methods are appropriate. The results support the conclusions. 

Round 2

Reviewer 1 Report

Thank you very much for the adaption of the manuscript. 

Author Response

Comment: Thank you very much for the adaption of the manuscript. 

Answer: We thank the reviewer for appreciating our efforts to improve the manuscript. His suggetions were very helpful and now we hope the paper could be consider for publication.